# All-optical nonequilibrium pathway to stabilising magnetic Weyl semimetals in pyrochlore iridates

Gabriel E. Topp[1], Nicolas Tancogne-Dejean [1], Alexander F. Kemper [2], Angel Rubio [1,3] & Michael A. Sentef[1]

Nonequilibrium many-body dynamics is becoming a central topic in condensed matter physics. Floquet topological states were suggested to emerge in photodressed bands under periodic laser driving. Here we propose a viable nonequilibrium route without requiring coherent Floquet states to reach the elusive magnetic Weyl semimetallic phase in pyrochlore iridates by ultrafast modification of the effective electron-electron interaction with short laser pulses. Combining ab initio calculations for a time-dependent self-consistent light-reduced Hubbard $U$ and nonequilibrium magnetism simulations for quantum quenches, we find dynamically modified magnetic order giving rise to transiently emerging Weyl cones that can be probed by time- and angle-resolved photoemission spectroscopy. Our work offers a unique and realistic pathway for nonequilibrium materials engineering beyond Floquet physics to create and sustain Weyl semimetals. This may lead to ultrafast, tens-of-femtoseconds switching protocols for light-engineered Berry curvature in combination with ultrafast magnetism.

---

[1] Max Planck Institute for the Structure and Dynamics of Matter, Center for Free Electron Laser Science, 22761 Hamburg, Germany. [2] Department of Physics, North Carolina State University, Raleigh 27695-8202 NC, USA. [3] Center for Computational Quantum Physics (CCQ), Flatiron Institute, 162 Fifth Avenue, New York, NY 10010, USA. Correspondence and requests for materials should be addressed to M.A.S. (email: michael.sentef@mpsd.mpg.de)

Ultrafast science offers the prospect of an all-optical design and femtosecond switching of magnetic and topological properties in quantum materials[1–6]. Tailored laser excitations offer the prospect of manipulating important couplings in solids, such as the magnetic exchange interaction[6,7] or electron–phonon coupling[8–11]. As a consequence light-triggered transitions between ordered and non-ordered collective phases can be observed, with dramatic changes of the electronic response on ultrafast time scales. Prominent examples are the switching to a hidden phase involving charge-density wave order[12], possible light-induced superconductivity in an organic material by phonon excitation[13], nonthermal melting of orbital order in a manganite by coherent vibrational excitation[14], or light-induced spin-density-wave order in an optically stimulated pnictide material[15]. Obviously, mutual electronic and lattice correlations are key to understanding the physics of such phase transitions both in and out of equilibrium. By contrast, many known topological materials are well understood by the Berry curvature of Bloch electrons in an effective single-quasiparticle picture. Similarly, the nonequilibrium extension to periodically driven Floquet states of matter[6,16–19] is in its simplest form well captured by driven noninteracting models[16]. From the point of view of pulsed laser engineering of novel states of matter, the drawback of Floquet states in weakly interacting systems is that any fundamental change induced by light in the material's band structure rapidly vanishes when the laser pulse is off. In that case, nonthermal distributions of quasiparticles in Floquet states do not critically affect the transient band structure, though they are relevant for response functions, for instance for describing the Floquet Hall effect[20]. This is distinctly different from the case of nonequilibrium ordered phases with an order parameter that crucially impacts the band structure itself. The slow and often nonthermal dynamics of driven correlated ordered phases[21,22], in particular in proximity to phase transitions, present the opportunity to affect band structures on time scales that are longer than the duration of the external perturbation. For instance, the critical slowing down of collective oscillations and relaxation dynamics is one of the hallmark signatures of nonthermal criticality[23] in interacting many-body systems out of equilibrium. As another recent example of ultrafast materials science in interacting systems, Floquet engineering of strongly correlated magnets into chiral spin states[24,25] has been proposed.

As a prototypical class of materials involving a striking combination of topology and magnetism, the 227 pyrochlore iridates were theoretically suggested as possible hosts of a variety of equilibrium phases[26,27] including antiferromagnetic insulating (AFI), as well as Weyl semimetallic (WSM) states with noncollinear all-in/all-out (AIAO) spin configurations. This topologically non-trivial phase is characterised by the appearance of pairs of linearly dispersing Weyl cones of opposite chirality. A phase transition between both phases, controlled by the size of the ordered magnetic moment and thus depending on Hubbard $U$, was conjectured to exist. However, convincing experimental evidence for the existence of the WSM phase in pyrochlore iridates has not been presented yet, with the majority of experiments pointing to AFI groundstates (cf. discussions in refs.[28–30]).

Here we propose to induce the elusive transition to the WSM phase by laser excitation. We use a combination of ab initio simulations of light-reduced Hubbard $U$ induced by an ultrashort laser pulse[31] and time-dependent magnetisation dynamics after a $U$ quench. We restrict our study to a model relevant for compounds with a non-magnetic R-site (R = Lu, Y, Eu), where the 5$d$ iridium electrons determine the magnetic order[27,32].

## Results

**Equilibrium properties of pyrochlore iridate model.** Figure 1a illustrates the key idea behind our work by presenting the low-temperature equilibrium phase diagram of a prototypical pyrochlore iridate model Hamiltonian[33],

$$H = \sum_{\mathbf{k}} \sum_{a,b} c_a^{\dagger}(\mathbf{k}) \left[ \mathcal{H}_{ab}^{NN}(\mathbf{k}) + \mathcal{H}_{ab}^{NNN}(\mathbf{k}) \right] c_b(\mathbf{k}) + H_U. \quad (1)$$

The indices $1 \leq a$, $b$, $c \leq 4$ run over the four sublattice sites. $\mathcal{H}_{ab}^{NN}$ and $\mathcal{H}_{ab}^{NNN}$ describe the bare nearest-neighbour (NN) and next-nearest-neighbour (NNN) hopping plus spin–orbit coupling, respectively. The local Hubbard repulsion $H_U = U \sum_{\mathbf{R}_i} n_{\mathbf{R}_i \uparrow} n_{\mathbf{R}_i \downarrow}$ is taken as the Hartree–Fock mean-field

$$H_U \rightarrow -U \sum_{\mathbf{k}a} \left( 2 \langle \mathbf{j}_a \rangle \cdot \mathbf{j}_a(\mathbf{k}) - \langle \mathbf{j}_a \rangle^2 \right), \quad (2)$$

where $\mathbf{j}_a(\mathbf{k}) = \sum_{\alpha\beta=\uparrow,\downarrow} c_{a\alpha}^{\dagger}(\mathbf{k}) \sigma_{\alpha\beta} c_{a\beta}(\mathbf{k})/2$ denotes the pseudospin operator. The groundstate phases within a mean-field approximation are a paramagnetic metal (PMM), an antiferromagnetic Weyl semimetal (WSM), and an antiferromagnetic insulator (AFI). The phase transitions are essentially controlled by the size of the magnetic order parameter, defined as the average length per unit cell of the pseudospin vectors in units of $\hbar \equiv 1$, $m \equiv \sum_a |\langle \mathbf{j}_a \rangle|/4$ (see Supplementary Fig. 1). This is in turn controlled by the hopping integrals between $\sigma$ orbitals, $t_\sigma$, and the local Coulomb repulsion, Hubbard $U$. In Fig. 1b we show the ordered magnetic moment, $m$, corresponding to the length of the magnetisation vector in the AIAO configuration as a function of $U$ for our choice of $t_\sigma = -0.62$ eV in equilibrium. Our choice of parameters is motivated by comparing with the size of the band gap from our density functional theory calculation to fix the energy units in the model, and we fix the ratio $t_\sigma/t_{\text{oxy}} = -0.775$, which controls the WSM–AFI transition to be of first order (cf. Fig. 1a), a worst-case scenario for switching across the transition as performed below. This first-order transition exhibits a small hysteresis region with a sharp change of $m$ near the transition. By contrast, the PMM–WSM transition at smaller $U$ is continuous. In the PMM phase, $m$ vanishes.

**Nonequilibrium driving.** In the following, we will make a case for nonequilibrium pathways to induce the elusive WSM state. Consistent with most of the experimental evidence, we choose an equilibrium initial state inside the AFI region (red square in Fig. 1), corresponding to $U = 1.28$ eV. This choice is not meant to be representative of a single specific compound but rather to generically represent to whole class of AFI pyrochlore iridates. Apparently, by controlling the strength of $U$ one might engineer a transition from the AFI to the WSM state. One way to effectively control $U$ in a correlated insulator was recently theoretically proposed[31], namely by a short laser excitation, employing ab initio time-dependent density functional theory plus dynamical $U$ (TDDFT + U) formalism[34]. Motivated by these results, we investigate here the influence of a short laser pulse on the effective interaction in insulating 227 iridates. We use TDDFT + U to show that a reduction of $U$ and of the magnetic order parameter $m$ can be induced in the pyrochlore iridates. We then take these first principles calculations to build a minimal intuitive model understanding of the nonequilibrium pathway to the light-induced WSM state by instantaneous quenches of $U$. This in-depth model investigation allows us to reveal the highly nonthermal character of the nonequilibrium ordered state, and to prove its WSM signatures by time-resolved photoemission spectroscopy calculations.

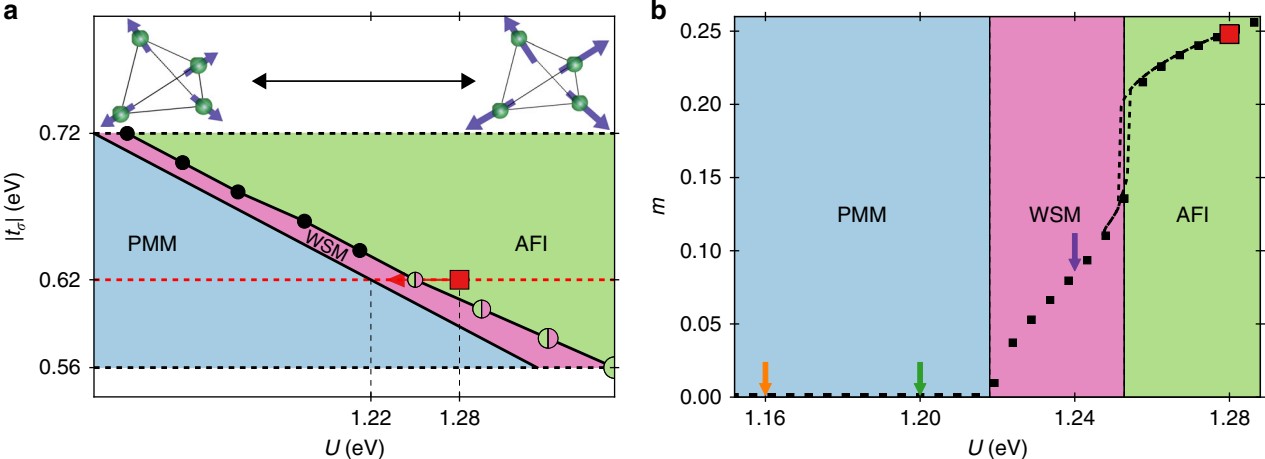

**Fig. 1** Equilibrium phases. **a** Phase diagram as a function of Hubbard $U$ and hopping integral, $t_\sigma$, at $T = 0.016$ eV (186 K). The red dashed line indicates the chosen parameter subspace for our calculations at $t_\sigma = -0.62$ eV. For small interactions, $U \leq 1.22$ eV, the system exhibits a paramagnetic metallic phase (PMM). In the intermediate regime, 1.22 eV $< U \leq 1.25$ eV, TRS is broken by spontaneous magnetic all-in/all-out (AIAO) order. A further increase of $U$ above 1.25 eV leads to a gapped antiferromagnetic insulating phase (AFI) with larger magnetisation. For $|t_\sigma| \geq 0.62$ eV the WSM–AFI transition is of second order, indicated by black dots. Below that value, the transition is of first order (bicoloured dots). The dots' increasing size corresponds to the increasing region of hysteresis. The red square indicates the chosen initial state ($U_i = 1.28$ eV) in parameter space. The red arrow shows the intended quenching direction. **b** Magnetisation as a function of $U$ for the choice of $t_\sigma = -0.62$ eV. The dashed lines indicate the region of hysteresis. The coloured arrows show the final values of $U = U_q$ right after the interaction quench for the nonequilibrium calculations (see Fig. 2)

In Fig. 2a we show the time evolution of $U_{eff}$ from TDDFT $+$ U for $Y_2Ir_2O_7$ driven by an ultrashort laser pulse, linearly polarised along the [001] direction, whose envelope is schematically indicated by the grey shaded area. Apparently, the effective Hubbard parameter is dynamically lowered by the light–matter interaction. This effect is explained by a dynamical enhancement of the electronic screening due to the delocalised nature of the pump-induced excited states[31]. The decrease of $U_{eff}$ follows the squared-sinusoidal pump envelope. After 20 fs $U_{eff}$ saturates at a finite value. The relative change of the effective interaction is controlled by the laser intensity. A higher intensity means a bigger and faster drop of the interaction. For the highest assumed realistic intensity in matter, $0.5 \times 10^{12}$ W cm$^{-2}$, $U_{eff}$ changes by 40%.

In a next step we investigate the real-time dynamics of the AIAO magnetic order in TDDFT $+$ U (Fig. 2b). First of all, the TDDFT groundstate calculation yields an AIAO magnetisation, $m = 0.23 \mu_B$, consistent with the model before laser excitation. Under the pump excitation, the magnetisation $m(t)$ is reduced from its equilibrium value. The reduction of $m(t)$ saturates as a function of pump intensity more quickly than the reduction of $U_{eff}$, while $m(t)$ still remains nonzero. This observation indicates that nonthermal effects are at play. We will get back to a more detailed discussion of nonthermality in the context of the model calculations below.

In the following, we employ the dynamical reduction of $U_{eff}$ in TDDFT $+$ U shown in Fig. 2a as input for model calculations with dynamical $U$ within the time-dependent self-consistent Hartree–Fock mean-field approximation. We include a dissipative coupling to a heat bath to mimic the openness of the electronic subsystem in the real material. To this end we couple the electrons to a Markovian fermionic heat bath giving rise to a Lindblad term (see Methods). The bath is kept at fixed temperature $T = 0.016$ eV (186 K), and we choose a system–bath coupling strength $\Gamma_0 = 0.008$ eV, corresponding to a characteristic time scale $\Gamma_0^{-1} \approx 80$ fs. Note that the microscopic details of the system–bath coupling are unimportant for the further discussion. The bath serves two main purposes here: (i) to thermalise the electronic subsystem as a whole to the bath

temperature, and (ii) to thermalise the electrons among each other through the bath.

The simplest and most extreme scenario to investigate is an instantaneous change of $U$ from its initial equilibrium value to quenched values $U_q$, which is a worst-case scenario for light-controlled phase transitions as it produces the strongest heating effects, as will be discussed below. In Fig. 2c we show the resulting time evolutions after $U$ quenches from the initial value to three different final values, corresponding to different laser pump intensities. We observe very fast changes of the magnetisation and a subsequent nonequilibrium state with reduced but nonzero magnetisation that slowly decays to its thermal value. In the following we focus on the characterisation of this nonthermal state at a fixed observation time $t_p = 123.4$ fs.

To further investigate this dynamically switched state which is in qualitative agreement with the TDDFT $+$ U results, we extract from the time-dependent total energies of the system computed with the Hartree–Fock Hamiltonian in the time-evolved state an effective electronic temperature corresponding to those energies (see Supplementary Fig. 3), cautioning that this effective temperature does *not* indicate thermalisation but is rather used as a means of talking about nonthermality in the following. The extracted values are placed in the equilibrium temperature-versus-$U$ phase diagram in Fig. 2d. While the $U_q = 1.24$ eV case still lies within the thermal magnetic phase region, this is clearly not the case for the stronger pulses reducing $U$ to $U_q = 1.20$ eV and $U_q = 1.16$ eV, respectively. For these quenched $U$ values, the thermal state has a vanishing magnetic order parameter in thermal equilibrium. For $U_q = 1.16$ eV there is no magnetic order even at zero temperature, while the dynamically induced state shows nonzero magnetic order even at a finite effective temperature. Without bath coupling, $\Gamma_0 = 0$, the effective nonequilibrium temperature increases quite rapidly for smaller quench values, $U_q$, as the increasing amount of additional energy pumped into the system is conserved. For the open system, the increase of temperature is much slower due to dissipation induced by the low-temperature heat bath. We take this as an indication that the observed nonequilibrium reduced but nonzero magnetic order is indeed nonthermal in character. This nonthermality makes Weyl

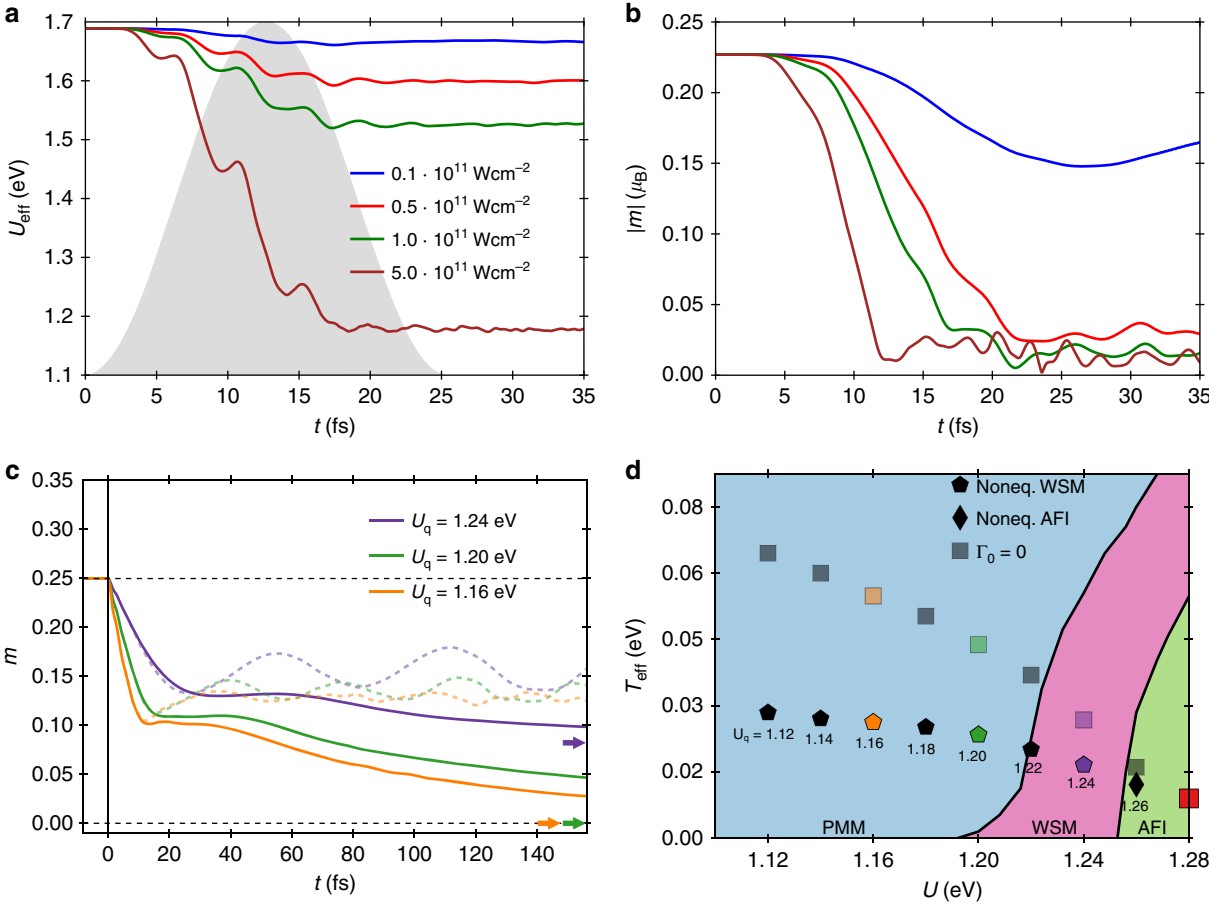

**Fig. 2** Optically induced nonthermal Weyl semimetal. **a** Self-consistent TDDFT + U calculation of $U_{eff}(t)$ (Ir 5d) for $Y_2Ir_2O_7$ under pump excitation (0.41 eV frequency, 25.4 fs duration). $U_{eff}$ decreases, initially following the pulse envelope. After 20 fs it reaches a plateau, which decreases for higher intensities. Relative changes of $U_{eff}$ are 1.2% (blue), 5.3% (green), 9.6% (red) and 40.4% (cyan). **b** Time-dependent AIAO magnetisation from TDDFT + U. For the smallest intensity (blue) the magnetisation drops by 25%. For all higher pump intensities, there is a massive decrease of the magnetisation by 88% (red) and 92% (green, brown), respectively. **c** Time-dependent AIAO magnetisation $m(t)$ (model) for bath coupling $\Gamma_0 = 0.008$ eV after instantaneous quenches at $t = 0$ from $U_i = 1.28$ eV to $U_q = 1.24$, 1.20 and 1.16 eV, respectively (dashed curves are without bath, $\Gamma_0 = 0$). Thermal $m$ values for $U_q$ are indicated by coloured arrows for reference. After a fast drop followed by an over-damped oscillation, the magnetisation slowly converges towards the respective thermal value. **d** Equilibrium phase diagram as a function of $T$ and $U$ with optically induced nonthermal states. Effective temperatures $T_{eff}$ (see Supplementary Fig. 3) of the nonequilibrium states at $t_p = 123.4$ fs after the quench from the initial thermal state (red square), for quench values $U_q$ as indicated (see Supplementary Fig. 2). Pentagons (squares) indicate nonequilibrium WSM states for the open system (closed system). (see Fig. 3). The black diamond indicates a nonequilibrium AFI state

states appear transiently in a much larger region of effective temperatures and $U$ values than they would in thermal equilibrium.

Importantly, such nonthermal order above the equilibrium critical temperature was observed for $U$ quenches in the antiferromagnetic phase of the single-band Hubbard model both in static and dynamical mean-field calculations[23]. The basic explanation for why the system does not thermalise rapidly is as follows: Integrability can block the system from thermalizing due to constants of motion that hinder relaxation to thermal values for some observables, such as the magnetisation. While our mean-field model in the absence of the heat bath falls into the integrable class, thermalisation is slowed down considerably even in nonintegrable systems close to a nonthermal critical point, at which the thermalisation time diverges. This is the case here for the second-order phase transition between the magnetic WSM and nonmagnetic metallic states. Therefore, nonthermality on time scales that exceed the time scale of the laser perturbation is not an artefact of the mean-field approximation employed in the

present work, but is expected to survive when true correlation effects are included.

**Time-resolved probing of Weyl fermions.** In order to reveal the WSM character of the pump-induced magnetically ordered state in the open system, we compute its time-resolved and angle-resolved photoemission spectroscopy (tr-ARPES) signal[18,35]. All information of the time-dependent electronic band structure is encoded in the double-time lesser Green's function, which in our case is calculated after the actual time propagation in a post-processing step by unitary double-time propagation of the initial density matrix (see Methods). The double-time lesser Green's function is the electronic propagator of a particle removed from the system at a point in time, and added back at a different time. Its Fourier transform from relative times to frequency gives the single-particle removal spectrum in equilibrium, of which the tr-ARPES signal is the time-dependent generalisation for the driven case. Thus the tr-ARPES photocurrent essentially monitors the occupied parts of the band structure.

The momentum-dependent and frequency-dependent photocurrent at time $t_p$ has an energy resolution given by the inverse of the probe duration. We use a probe duration $\sigma_p = 20.6$ fs unless denoted otherwise, leading to a broadening of the spectral lines proportional to the inverse of the probe duration due to energy–time uncertainty. For reference, in Fig. 3a, we show the ARPES signal for the equilibrium band structure of the initial state ($U_i = 1.28$ eV) along a high-symmetry path in the first Brillouin zone. The AFI phase is clearly identified by the separation of the occupied valence bands from the empty conduction bands by an energy gap, $E_G \approx 0.15$ eV, at the $L$ point. The chemical potential $\mu = 0$ lies within that gap.

In Fig. 3b–d we display computed time-resolved band structure of the nonequilibrium steady-state at probe time $t_p = 123.4$ fs, after quenches as in Fig. 2b. We find Weyl cones in all cases, indicated by the coloured arrows. As the magnetic order is practically constant during the probe duration, the photocurrent nicely matches the instantaneous eigenenergy spectrum of the Hamiltonian, depicted by the black solid lines. For increasing quench sizes, the Weyl cone moves along $\Gamma - L$ towards $\Gamma$.

**Lifetime and relaxation of Weyl semimetal.** Finally, we address the question of the minimum lifetime of the nonthermal WSM state. In reality, the light-induced reduction of $U$ will not persist indefinitely due to coupling of the electronic system to the environment, for instance to lattice vibrations, which are not included in our TDDFT + U simulations. Electron–phonon relaxation is always present in materials and typically has associated time scales of hundreds of femtoseconds up to picoseconds, usually somewhat but not much slower than typical pump pulse durations of tens to hundreds of femtoseconds[36]. Therefore, we take here a worst-case scenario and assume that $U$ is reduced only on a time scale that is comparable to the dissipative time scale introduced through the heat-bath coupling. We therefore investigate the dynamics of the magnetisation and the nonequilibrium Weyl cone for a time-dependent $U(t)$ shown in Fig. 4a. Here the minimum in $U(t)$ agrees with the smallest value $U_q$ used for the instantaneous quenches (see Fig. 2b). Figure 4b shows the corresponding time-dependent magnetisation, which closely follows $U(t)$, for the same heat bath as employed before. The magnetisation $m(t)$ reaches a smaller minimum than in the quenched case, which reflects the fact that the magnetisation follows more closely the expected thermal value corresponding to $U(t)$ when $U(t)$ changes more slowly than in the quench case. At the end of the $U$ modulation, the magnetisation is not fully relaxed back to its initial thermal value but slowly decays back to it due to coupling to the heat bath.

Figure 4c shows the calculated ARPES spectrum of the inititial state before the $U$ modulation. At $t_p = 20$ fs (Fig. 4d), the reduced magnetisation leads to a reduced band gap compared to the initial state. At $t_p = 50$ fs (Fig. 4e) the system undergoes a rapid change in magnetisation, resulting in more diffuse bands. Around $t_p = 100$ fs (Fig. 4f) $m(t)$ reaches a minimum and changes more slowly than at earlier times. The Weyl cone emerging between L and $\Gamma$ shows that the system has reached a WSM state. The tr-ARPES bands nicely match the instantaneous bands. Afterwards, the relaxation of the system is reflected in the shift of the Weyl cone back towards $\Gamma$ at $t_p = 150$ fs (Fig. 4g). Importantly the WSM state still persists outside the FWHM of $U(t)$ indicating that interaction-induced nonthermal states of matter can live longer than the duration of the pump laser pulse, in contrast to Floquet states in quasi-noninteracting systems. This longevity is directly related to the fact that the dynamics of $m(t)$ are slower than the dynamics of $U(t)$. Figure 4h shows the tr-ARPES signal after $U(t)$ has completely relaxed. The system is again found in the AFI

phase with a still slightly reduced gap compared to the initial thermal state.

**Discussion**

In summary, we propose a robust and efficient novel ultrafast route towards light-induced topology via dynamical modulation of Hubbard $U$ for nonequilibrium materials engineering as an alternative to Floquet engineering in quasi-noninteracting band structures. In our simulations, the laser-induced or quench-induced excitation leads to a dynamical reduction of magnetic order inducing an ultrafast transition to the Weyl semimetallic phase in pyrochlore iridates. The range of effective electronic temperatures in which nonzero reduced magnetisation is obtained is larger than expected for quasi-thermal states, highlighting that nonthermal order might allow to avoid finetuning in the quest for the light-induced magnetic Weyl phase. Importantly, the nonthermal character of transient states on long time scales has been shown to be a generic feature of ordered phases in proximity to critical points due to prethermalization even beyond the static mean-field approximation[23]. Crucially, the appearance of nonthermal order is not a prerequisite for our proposal to work at all. The time scales for nonthermal order mainly set a window of lifetimes and pump fluences under which light-induced magnetic Weyl states become observable experimentally. In practice, these windows will be restricted mainly by the relaxation dynamics of the dynamically modulated $U$ and magnetisation due to coupling to phonons.

Overall our results imply that ultrafast pathways are a promising route to reach the elusive topological Weyl semimetallic state in pyrochlore iridates. We suggest time-resolved and angle-resolved photoemission spectroscopy as a means of probing the dynamically generated Weyl cones. Specifically for chiral Weyl fermions, such emergent states could also be probed all-optically in a time-domain extension of the recently demonstrated static photocurrent response to circularly polarised light[37].

In a broader context our combined study of ab initio TDDFT + U under explicit laser excitation and model dynamics under interaction quenches furthermore bridges the so-far largely disconnected fields of laser-driven phenomena[38,39] on the one hand and thermalisation after quenches[40–43] on the other hand. Moreover the combination of light-controlled interactions and interaction-controlled topology adds to the variety of control knobs in the time domain to dynamically engineer topological phases of matter in solids on ultrafast time scales.

**Methods**

**Pyrochlore Hamiltonian.** The Pyrochlore structure is given by an fcc Bravais lattice with a four-atomic basis. We define the lattice vectors

$$\mathbf{b}_1 = (0, 0, 0), \quad \mathbf{b}_2 = (0, 1, 1),$$

$$\mathbf{b}_3 = (1, 0, 1), \quad \mathbf{b}_4 = (1, 1, 0), \tag{3}$$

with a lattice constant $a = 4$. We work with the time-reversal invariant Hamiltonian introduced in ref.[33], which, by assuming a single Kramers doublet at each Ir site, results in an eight-band model including splitting of the partially filled $5d$ electron shell due to spin–orbit coupling. In momentum space, the Hamiltonian takes the form

$$H = \sum_{\mathbf{k}} \sum_{a,b} c_a^\dagger(\mathbf{k}) \left[ \mathcal{H}_{ab}^{NN}(\mathbf{k}) + \mathcal{H}_{ab}^{NNN}(\mathbf{k}) \right] c_b(\mathbf{k}) + H_U,$$

with

$$\mathcal{H}_{ab}^{NN}(\mathbf{k}) = 2(t_1 + t_2 i\sigma \cdot \mathbf{d}_{ab}) \cos[\mathbf{k} \cdot \mathbf{b}_{ab}],$$
$$\mathcal{H}_{ab}^{NNN}(\mathbf{k}) = \sum_{\mathbf{c} \neq \mathbf{a}, \mathbf{b}} \left\{ t_1'(1 - \delta_{ab}) + i\sigma \cdot [t_2'(\mathbf{b}_{ac} \times \mathbf{b}_{cb}) \right.$$
$$\left. + t_3'(\mathbf{d}_{ac} \times \mathbf{d}_{cb})] \right\} \cos[\mathbf{k} \cdot (-\mathbf{b}_{ac} + \mathbf{b}_{cb})]. \tag{4}$$

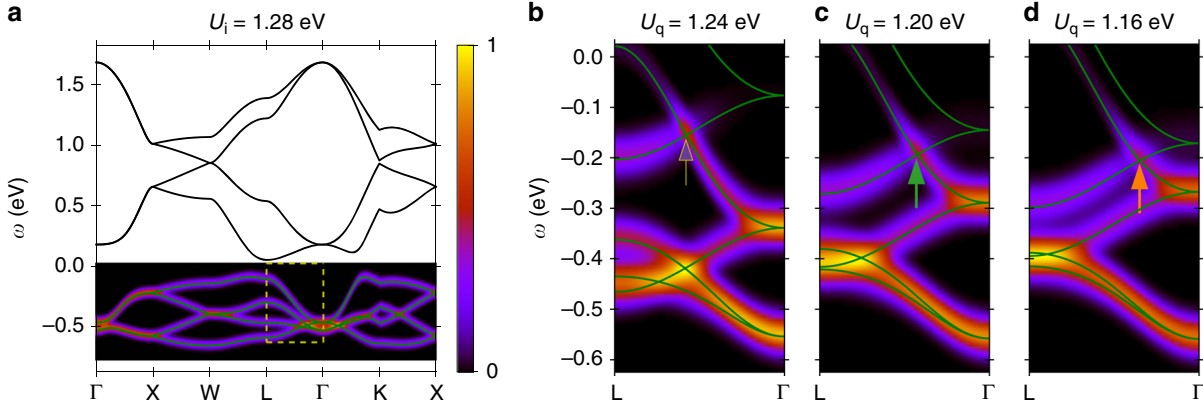

**Fig. 3** Time-resolved ARPES detection of Weyl fermions. **a** Calculated photocurrent from tr-ARPES of the equilibrium band structure along high-symmetry path for an initial AFI state ($U_i$ = 1.28 eV, $t_\sigma = -0.62$ eV) at $T$ = 0.016 eV. The conduction bands are separated from the valence bands by an energy gap $E_G \approx 0.15$ eV (AFI) at the $L$ point. The yellow square highlights the region of interest. **b–d** tr-ARPES band structure along the high-symmetry line $L - \Gamma$ at probe time $t_p$ = 123.4 fs. The quenched interaction values, $U_q$ = 1.24, 1.20 and 1.16 eV, correspond to increasing $\Delta U$ between initial and quenched $U$ and thus stronger driving fields. The black solid lines show the instantaneous band structures of the Hamiltonian at $t = t_p$. The coloured arrows indicate the dynamically generated Weyl points. The Weyl cones are shifted along the high symmetry line towards the $\Gamma$-point with increasing $\Delta U$

Here, the indices $1 \le a, b, c \le 4$ run over sublattices. The operator $c_a(\mathbf{k})^\dagger = \left(c_{a\uparrow}(\mathbf{k})^\dagger, c_{a\downarrow}(\mathbf{k})^\dagger\right)$ is a spinor, where the second lower index refers to a global pseudospin-1/2 degree of freedom. The matrices $\mathcal{H}_{ab}^{NN}$ and $\mathcal{H}_{ab}^{NNN}$ describe the nearest-neighbour (NN) and next-nearest-neighbour (NNN) hopping including spin–orbit coupling, respectively. The vector $\sigma = (\sigma_x, \sigma_y, \sigma_z)$ contains the Pauli matrices. The following real-space vectors appear in the Hamiltonian:

$$\mathbf{d}_{ij} = 2\mathbf{a}_{ij} \times \mathbf{b}_{ij}, \tag{5}$$

$$\mathbf{a}_{ij} = \frac{1}{2}\left(\mathbf{b}_i + \mathbf{b}_j\right) - (1,1,1)/2, \tag{6}$$

$$\mathbf{b}_{ij} = \mathbf{b}_j - \mathbf{b}_i. \tag{7}$$

Starting from a localised Hubbard repulsion of the form $H_U = U \sum_{\mathbf{R}_i} n_{\mathbf{R}_i \uparrow} n_{\mathbf{R}_i \downarrow}$, we use the Hartree–Fock mean-field decoupling

$$H_U \rightarrow -U \sum_{\mathbf{k}i} \left(2\langle \mathbf{j}_i \rangle \cdot \mathbf{j}_i(\mathbf{k}) - \langle \mathbf{j}_i \rangle^2\right), \tag{8}$$

where $\mathbf{j}_i(\mathbf{k}) = \sum_{\alpha\beta=\uparrow,\downarrow} c_{i\alpha}^\dagger(\mathbf{k}) \sigma_{\alpha\beta} c_{i\beta}(\mathbf{k})/2$ denotes the pseudospin of band $i = 1, ..., 4$ and $\langle \mathbf{j}_i \rangle = \frac{1}{V} \sum_{\mathbf{k}} \mathbf{j}_i(\mathbf{k})$ its mean expectation value. Here, $\mathbf{k}$ runs over the $k$-space volume of the first Brillouin zone $V$. We sample the BZ with a grid of $30 \times 30 \times 30$ points for the full BZ in $k$-space. For our calculations we use a half-volume reduced BZ by exploiting inversion symmetry.

**Microscopic parameters.** Following ref. [33] we use

$$t_1 = \frac{130 t_{oxy}}{243} + \frac{17 t_\sigma}{324} - \frac{79 t_\pi}{243}, \quad t_1' = \frac{233 t_{\sigma'}}{2916} - \frac{407 t_\pi'}{2187},$$
$$t_2 = \frac{28 t_{oxy}}{243} + \frac{15 t_\sigma}{243} - \frac{40 t_\pi}{243}, \quad t_2' = \frac{t_\sigma'}{1458} - \frac{220 t_\pi'}{2187}, \tag{9}$$
$$t_3' = \frac{25 t_\sigma'}{1458} + \frac{460 t_\pi'}{2187},$$

where $t_{oxy}$ and $t_\sigma$, $t_\pi$ denote the oxygen-mediated and direct-overlap NN hopping, respectively. The dashed parameters refer to the NNN hopping. We choose $t_\pi = -2t_\sigma/3$, $t_\sigma'/t_\sigma = t_\pi'/t_\pi = 0.08$. If not denoted otherwise, we work in units of fs (time) and eV (energy). A comparison between the ground-state band structures obtained from DFT calculations and model calculations yields an oxygen-mediated hopping of approximately $t_{oxy} = 0.8$ eV.

**Self-consistent thermal state.** In the mean-field decoupled Hubbard term Eq. (8) the $\langle \mathbf{j}_i \rangle$ are proportional to the spontaneous magnetic moments of the 5d electrons and thus determine the overall magnetic configuration. We define the magnetic order parameter as the average length of the pseudospin vector

$$m \equiv \frac{1}{4}\sum_a |\langle \mathbf{j}_a \rangle| = \frac{1}{4}\sum_a \sqrt{\langle j_a^x \rangle^2 + \langle j_a^y \rangle^2 + \langle j_a^z \rangle^2}, \tag{10}$$

which is proportional to the magnetic moment per unit cell, carried by the 5d

iridium electrons (in units of $\hbar \equiv 1$). Assuming half-filling, we keep the total number of electrons per unit cell, $n_{target} = 4$, constant. In order to determine the thermal initial state configuration for a fixed temperature, $\beta^{-1}$, we calculate the initial values of the magnetic moments and the chemical potential, $\mu$, in a self-consistent procedure. To this end we adjust the chemical potential accordingly. This procedure is repeated until a converged magnetic order parameter $|m_{n+1} - m_n| < 10^{-15}$ is reached in consecutive iterations of the self-consistency loop.

**Unitary time propagation.** For a closed system the dynamics of the density operator and thus the time-dependent magnetisation are governed by the von-Neumann equation, $\dot{\rho}(t) = -i[H(t), \rho(t)]$. We use the two-step Adams predictor-corrector method, a linear multi-step scheme, for its numerical solution. First, the explicit two-step Adams–Bashforth method to calculate a prediction of the value at the next time step is used. This takes the form

$$\begin{aligned} \rho(t) &= \rho(t - \Delta t) - i\Delta t[H(t - \Delta t), \rho(t - \Delta t)], \\ \rho_{pred}(t + \Delta t) &= \rho(t) - i\tfrac{3\Delta t}{2}[H(t), \rho(t)] \\ &\quad + i\tfrac{\Delta t}{2}[H(t - \Delta t), \rho(t - \Delta t)]. \end{aligned} \tag{11}$$

The first line's Euler step is only needed to calculate the very first time step. Afterwards, the implicit Adams–Moulton corrector is applied

$$\begin{aligned} \rho_{corr}(t + \Delta t) = \rho(t) - i\tfrac{\Delta t}{2}\Big(&\big[H(t + \Delta t), \rho_{pred}(t + \Delta t)\big] \\ &+ [H(t), \rho(t)]\Big). \end{aligned} \tag{12}$$

We typically use 200,000–400,000 time steps for the propagation. This corresponds to a smallest step size of $\Delta t = 0.0005$ fs. Convergence in the step size is always ensured.

**Non-unitary time evolution.** We introduce relaxation processes by coupling the electronic pyrochlore system to thermal fermionic reservoirs. The non-unitary dynamics of the reduced system's degrees of freedom are described in the framework of system-bath theory by a Lindblad master equation. We choose the following form for the Lindblad dissipator for a time-independent Hamiltonian:

$$\mathcal{D}(\rho_S) = \sum_\omega \sum_{\alpha\beta} \gamma_{\alpha\beta}(\omega)\left(A_\beta(\omega)\rho_S A_\alpha^\dagger - \frac{1}{2}\left\{A_\alpha^\dagger(\omega)A_\beta(\omega), \rho_S\right\}\right). \tag{13}$$

Here, $A_\alpha(\omega)$ describe system coupling operators of a general interaction Hamiltonian, $H_I = \sum_\alpha A_\alpha \otimes B_\alpha$, expanded in the energy eigenbasis of the system Hamiltonian. As the system under consideration and thus its Hamiltonian is time-dependent, the instantaneous eigenbasis approximation[44] is carried out. Here, the coupling operators take the form

$$\begin{aligned} A_\alpha(\omega) &= \sum_{\epsilon_b - \epsilon_a = \omega} |\epsilon_a\rangle\langle\epsilon_a|A_\alpha|\epsilon_b\rangle\langle\epsilon_b| \\ &\rightarrow A_\alpha(\omega(t)) = \sum_{\epsilon_b(t) - \epsilon_a(t) = \omega(t)} |\epsilon_a(t)\rangle\langle\epsilon_a(t)|A_\alpha|\epsilon_b(t)\rangle\langle\epsilon_b(t)|, \end{aligned} \tag{14}$$

where $H_S(t)|\epsilon_a(t)\rangle = \epsilon_a(t)|\epsilon_a(t)\rangle$ denotes the instantaneous energy eigenbasis of

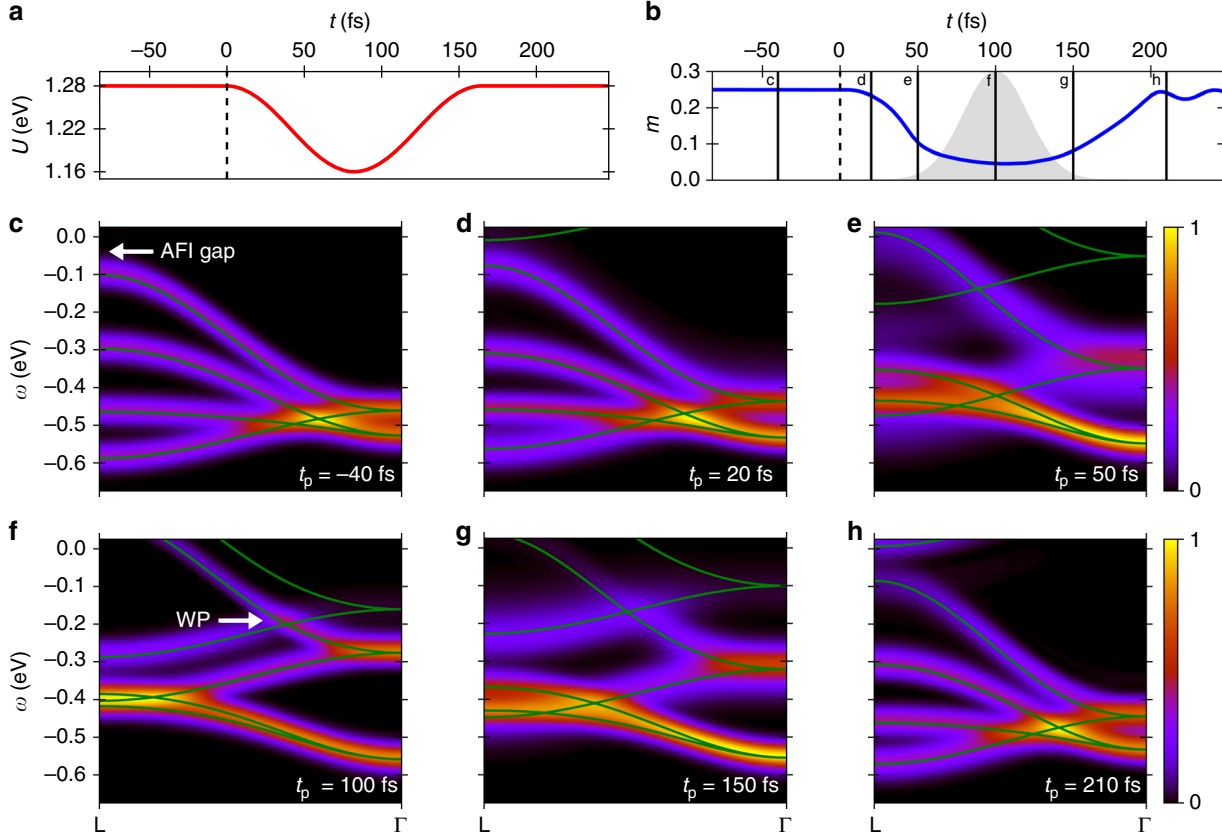

**Fig. 4** Relaxation of nonthermal Weyl semimetal. **a** Time-dependent interaction relaxing back to the initial value, $U(0 \leq t \leq 150 \text{ fs}) = U_i - (U_i - U_q) \sin(\pi \cdot t/150 \text{ fs})$. **b** Corresponding time evolution of the magnetisation. As in the instantaneous case, the magnetisation is initially reduced before slowly relaxing back towards its initial value. The vertical solid lines indicate the probe times $t_p$ for which the tr-ARPES signals are shown in **c–h**, with a probe pulse duration $\sigma_p = 20.6$ fs indicated by the grey-shaded Gaussian envelope for one selected probe time. **c–h** Tr-ARPES signals at different probe times (indicated in **b**) along $L - \Gamma$. The solid lines correspond to the effective band structures of the time-dependent Hamiltonian averaged over the FWHM of the probe pulse

the system Hamiltonian. The Hermitian coefficient matrix $\gamma_{\alpha\beta}$ is defined by the Fourier transform of the bath correlation functions $\int_{-\infty}^{+\infty} ds e^{i\omega(t)s} \langle B_\alpha^\dagger(s) B_\beta(0) \rangle$.

For every band $a$ and momentum $k$ we define two system coupling operators $A_1^{a,k} \equiv d_{a,k}(t)$ and $A_2^{a,k} \equiv d_{a,k}^\dagger(t)$, which annihilate and create a quasi-particle with the instantaneous band energy $\epsilon_{a,k}(t)$, respectively. Accordingly, we define two bath coupling operators $B_1^{a,k} \equiv \sum_n t_n^{a,k}(b_n^{a,k})^\dagger$ and $B_2^{a,k} \equiv \sum_n (t_n^{a,k})^* b_n$, where $t_n^{a,k}$ denotes the transition matrix element and $n$ the mode index. Assuming the bath to be in a thermal steady state and momentum-independent transition rates $\Gamma_{a,k}(\omega) \equiv \sum_n |t_n^{a,k}|^2 \delta(\omega - \omega_n^{a,k}) \to \Gamma_0$ in the wide-band limit, this introduces relaxation dynamics of the form

$$\frac{d}{dt} \langle d_{a,k}^\dagger(t) d_{b,k}(t) \rangle_t = \text{unitary evolution}$$
$$- 2\Gamma_0 \left[ \langle d_{a,k}^\dagger(t) d_{b,k}(t) \rangle_t - n_F(\epsilon_{a,k}(t), \mu(t)) \right] \delta_{ab} \quad (15)$$
$$- 2\Gamma_0 \langle d_{a,k}^\dagger(t) d_{b,k}(t) \rangle_t (1 - \delta_{ab}).$$

In the above equation $n_F(\epsilon, \mu) \equiv (1 + \exp(\beta(\epsilon - \mu)))^{-1}$ denotes the Fermi–Dirac distribution at inverse temperature $\beta = (k_B T)^{-1}$. Throughout the paper we use eV units for temperature, setting $k_B \equiv 1$ and employing that 1 eV corresponds to 11,605 K. Importantly, the chemical potential is time-dependently adjusted, $\mu \to \mu(t)$, in order to keep the particle number in the system constant. The first non-unitary term on the right-hand side of the above equation leads to thermalization of the occupations, the second term induces an exponential decay of the interband transitions. The relaxation time scale is governed by the inverse coupling $\Gamma_0^{-1} \approx 80$ fs. We use a bath temperature $T = 0.016$ eV (186 K).

**Time-resolved ARPES.** In order to monitor the nonequilibrium changes in the electronic band structure in a time-resolved fashion, we use a theoretical

implementation of a time-resolved angle-resolved photoemission spectroscopy (tr-ARPES) probe measurement. The central mathematical object, in order to calculate the observable photocurrent, is the two-times lesser Green's function

$$G_{ab}^<(\mathbf{k}, t_r + t, t_r + t') \equiv i \sum_{\sigma\sigma'} \langle c_a^\dagger(\mathbf{k}, t_r + t) c_b(\mathbf{k}, t_r + t') \rangle,$$
$$\approx U(\mathbf{k}, t, t_r) G_{ab}^<(\mathbf{k}, t_r, t_r) U(\mathbf{k}, t', t_r)^\dagger, \quad (16)$$
$$G_{ab}^<(\mathbf{k}, t_r, t_r) = i\rho_S(t_r).$$

The full lesser Green's function of the non-unitary time evolution is approximated by a unitary two-time propagation of the time-dependent density matrix from all real time steps $t_r$. The general unitary propagator is of the form

$$U(\mathbf{k}, t, t') = \mathcal{T} \exp\left[ -i \int_{t'}^t ds H(\mathbf{k}, s) \right]. \quad (17)$$

Using the discretization of the real-time axis and respecting the time-ordering this can be approximated as

$$U(\mathbf{k}, t, t') \approx \Pi_{j=1}^{N_{t,t'}} \exp[-iH(\mathbf{k}, t - j\Delta t/2)\Delta t], \quad (18)$$

where $N_{t,t'}$ denotes the number of time steps between $t$ and $t'$. For the post-processing calculation of $G^<$ we use a smaller time stepping, $\Delta t = 0.01$ fs.

The photocurrent can be computed by[35]

$$I(\mathbf{k}, \omega, \Delta t) = \text{Im} \int dt \int dt' s_{\sigma_p}(t_p - t) s_{\sigma_p}(t_p - t') \exp[i\omega(t - t')] \text{Tr}\{G^<(\mathbf{k}, t, t')\}.$$

Here, $t_p$ refers to the centre time of the probe pulse, $\omega$ to its frequency. Here $s_{\sigma_p}(t)$ defines the probe pulse duration via the width of the Gaussian-shaped probe pulse envelope in both directions in time.

**TDDFT + U simulations**. The evolution of the spinor states and the evaluation of the time-dependent Hubbard $U$ and magnetisation are computed by propagating the generalised Kohn–Sham equations within time-dependent density functional theory including mean-field interactions (TDDFT + U), as provided by the Octopus package[34,45], using the ACBN0 functional[46] together with the local-density approximation (LDA) functional for describing the semilocal DFT part. We compute ab initio the Hubbard $U$ and Hund's $J$ for the $5d$ orbitals of iridium. We employ norm-conserving HGH pseudopotentials[47], a Fermi–Dirac smearing of 25 meV, a real-space grid spacing of 0.3 atomic units, and an $8 \times 8 \times 8$ k-point grid to get converged results of $U$. We find that the inclusion of semi-core states of yttrium and iridium are important for obtaining accurate band structures; thus valence electrons explicitly included are Y: $4s^2$, $4p^6$, $4d^1$ and $5s^2$; and Ir: $5s^2$, $5p^6$, $5d^7$ and $6s^2$.

All TDDFT calculations for bulk $Y_2Ir_2O_7$ (spacegroup $Fd3m$, number 227) were performed using the primitive cell containing 22 atoms, without a priori assuming symmetries in order to obtain the correct magnetic ground-state. We employ the experimental values for the lattice parameter and atomic positions[48]. Starting from a random magnetic configuration as an initial guess, we find the all-in all-out configuration at the end of the self-consistent ground-state calculation (see Supplementary Fig. 4), in agreement with the model results and previous studies[27,33]. Very similar values for the magnetic moments for the Ir atoms are obtained from the spherical averaging of the total electronic density or from the density matrix of the localized orbitals (entering in the evaluation of the ACBN0 functional), demonstrating that the magnetic properties arise from the localized orbitals. We obtain the groundstate ab initio values of $U = 2.440$ eV and $J = 0.74$ eV, leading to an effective $U_{eff} = U - J = 1.69$ eV.

For the time-dependent simulations, the laser is coupled to the electronic degrees of freedom via the standard minimal coupling prescription using a time-dependent, spatially homogeneous vector potential $\mathbf{A}(t)$, with electric field $\mathbf{E}(t) = -\frac{1}{c}\frac{\partial \mathbf{A}(t)}{\partial t}$. We consider a laser pulse of 12.7 fs duration at full-width half-maximum (FWHM) with a sin-square envelope corresponding to a total width of 25.4 fs, and the carrier wavelength $\lambda$ is 3000 nm, corresponding to $\omega = 0.41$ eV. We choose the driving field to be linearly polarised along the [001] direction (c-axis in Fig. 4). In all our calculations, we used a carrier-envelope phase of $\phi = 0$.

## Data availability

All data generated and analysed during this study are available from the corresponding author upon reasonable request.

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

## Acknowledgements

We thank C. Timm for suggesting the pyrochlore iridates as a candidate class of materials to light-induced nontrivial topology beyond Floquet states. Discussions with A. de la Torre and D. Kennes are gratefully acknowledged. G.E.T. and M.A.S. acknowledge financial support by the DFG through the Emmy Noether programme (SE 2558/2-1). A.R. and N.T.-D. acknowledge financial support by the European Research Council (ERC-2015-AdG-694097) and European Union's H2020 programme under GA no. 676580 (NOMAD).

## Author contributions

G.E.T. performed the model calculations. N.T.-D. performed the TDDFT + U calculations. The project was conceived by M.A.S., A.F.K., and A.R. All authors discussed the results. G.E.T. and M.A.S. wrote the manuscript, with feedback from N.T.-D., A.F.K. and A.R.

## Additional information

**Competing interests:** The authors declare no competing interests.

