## [Peer Review file · Nature Communications]

Reviewers' comments:

Reviewer #1 (Remarks to the Author):

The manuscript "All-optical nonthermal pathway to stabilizing magnetic Weyl semimetals in pyrochlore iridates" is a theoretical work that proposes a non-equilibrium route to realizing a Weyl semimetal. More specifically, the authors describe how an appropriate laser pulse can lead to a transient Weyl semimetal in a family of Iridium-based oxides, the pyrochlore iridates. The numerical analysis relies on model Hamiltonians and ab initio methods. The resulting Weyl fermions could be experimentally probed using time- and angle-resolved photoemission spectroscopy.

The pyrochlore iridates constitute a family of materials that has been at the forefront of condensed matter research in the past few years. These materials combine strong spin orbit coupling with strong electron-electron interactions. Experiments have observed non-collinear magnetic order, metal-insulator transitions, non-trivial domain-wall states. Some experiments have seen signatures of Weyl fermions under hydrostatic pressure, but more data would be needed to reach a definite conclusion. In their manuscript, the Topp et al suggest a new route for a Weyl semimetal in these materials. They propose that a non-equilibrium pump experiment can be used to induce a Weyl phase in a material that's in a conventional equilibrium phase. This combines three key research directions: topological phases, electronic correlations, and non-equilibrium phenomena. Non-equilibrium quantum systems in 3D are very difficult to study, and drastic approximations generally need to be made. This is the case in the present manuscript. To the credit of the authors, they also use ab initio methods (with the LDA approximation) to bring the analysis closer to experiments. However, one is still left with doubts as to whether the predictions can be trusted. This and other limitations prevent me from suggesting publication in Nature Communications. Below I give more detailed comments.

One of the main claims of the paper is that the laser pulse leads to a non-thermal (Weyl) state that has a relatively long lifetime. However, the authors do not explain why the system does not thermalize. In interacting quantum systems, thermalization can occur following a quench. Here, the non-thermal nature of the long-lived state is unclear. One can worry it is a relic of the approximations made. For instance, it is known that non-interacting (e.g. mean field states) do not thermalize.

- Weyl semimetals have been observed experimentally in equilibrium, where they are easier to probe. Obtaining them via a non-equilibrium pathway is thus not as exciting. Combined with the fact that time-resolved ARPES measurements will be difficult, I do not expect that many experimentalists will attempt to follow this numerical recipe. If the non-equilibrium state had intrinsically new properties, it would be more exciting, but this is not discussed in the manuscript.

- In the numerical simulations, how is the laser pulse coupled to the electronic degrees of freedom? Also, can the authors explain the choice of polarization? Would a circularly polarized beam lead to significant differences?

- Why is the AFI phase referred to as a Mott phase? - The double-time lesser Green's function is mentioned without explanation in the text. This is not appropriate for the broad readership of Nature Communications. The authors should add a heuristic explanation.

- On page 4, the authors refer to a DFT calculation for the band gap, but not provide a reference.

- In Fig. 3, what causes the broadening of the spectral lines?

- On page 2, the authors use reference 11 to mention interesting equilibrium phases predicted for the

pyrochlore iridates. However, the following reference has appeared prior to 11:

Pesin & Balents, Mott physics and band topology in materials with strong spin-orbit interaction, Nature Physics volume 6, pages 376–381 (2010)

- On page 3, references 26-27 are used regarding the origin of magnet order in the iridates. The earlier reference 11 seems more appropriate, as well as the subsequent review article: Witczak-Krempa, Chen, Kim, Balents, Correlated quantum phenomena in the strong spin-orbit regime, Annual Review of Condensed Matter Physics, Vol. 5: 57-82 (2014)
This review seems to also summarize the physics encoded in the Hubbard model used by the authors.

Reviewer #2 (Remarks to the Author):

This manuscript theoretically proposes a way to realize the Weyl semimetal along with related magnetism in pyrochlore iridates. For that the authors evoke a ultrafast, nonequilibrium modification of the electron interaction by laser pulses. The proposal is interesting as a nonequilibrium modification of material properties with the iridates being of currently much studied one. So I basically recommend its publication, but I have several comments.

1. The authors heavily rely on the (DFT+U) formalism. For a description of real materials the method is often employed, but at the same time it is well known that the method has some ambiguity due to double counting terms. So the authors should discuss how this may affect the treatment, along with its relevance to t-dependent (relaxation) phenomena. Alternative method could be t-dependent DMFT, and they might like to comment on this, too.

2. The authors say that laser-modified states in weakly-interacting systems are distinct from those for ordered phases for correlated electrons with slow and often nonthermal dynamics[22,23]. However, already in the original proposal of the Floquet topological insulator[7] and subsequent papers [Dehghani et al, PRB 91, 155422 (2015)], nonthermal distribution in Floquet states are shown to be important, so the statement can be misleading. Ref[22], by the way, treats an electron-phonon system where phonon relaxation can be involved.

Rests are some points on citation.

3. When the authors say that "Floquet topological states" ... captured by driven noninteracting models, ref[7] may be cited as a first example.

4. The authors stress the distinction between the existing Floquet states and the present proposal of the electron interaction modification. That is a good point, but still they can refer to eg Floquet engineering of strongly-correlated magnets into chiral spin states [Kitamura et al, PRB 96, 014406 (2017)] as a related subject.

Reviewers' comments:

Reviewer #1 (Remarks to the Author):

The manuscript “All-optical nonthermal pathway to stabilizing magnetic Weyl semimetals in pyrochlore iridates” is a theoretical work that proposes a non-equilibrium route to realizing a Weyl semimetal. More specifically, the authors describe how an appropriate laser pulse can lead to a transient Weyl semimetal in a family of Iridium-based oxides, the pyrochlore iridates. The numerical analysis relies on model Hamiltonians and ab initio methods. The resulting Weyl fermions could be experimentally probed using time- and angle-resolved photoemission spectroscopy.

The pyrochlore iridates constitute a family of materials that has been at the forefront of condensed matter research in the past few years. These materials combine strong spin orbit coupling with strong electron-electron interactions. Experiments have observed non-collinear magnetic order, metal-insulator transitions, non-trivial domain-wall states. Some experiments have seen signatures of Weyl fermions under hydrostatic pressure, but more data would be needed to reach a definite conclusion.

In their manuscript, the Topp et al suggest a new route for a Weyl semimetal in these materials. They propose that a non-equilibrium pump experiment can be used to induce a

Weyl phase in a material that's in a conventional equilibrium phase. This combines three key research directions: topological phases, electronic correlations, and non-equilibrium phenomena. Non-equilibrium quantum systems in 3D are very difficult to study, and drastic approximations generally need to be made. This is the case in the present manuscript. To the credit of the authors, they also use ab initio methods (with the LDA approximation) to bring the analysis closer to experiments. However, one is still left with doubts as to whether the predictions can be trusted. This and other limitations prevent me from suggesting publication in Nature Communications. Below I give more detailed comments.

We appreciate the reviewer's careful reading of our manuscript and his/her assessment of the merits of our approach. We hope that the reviewer appreciates the new ideas and positive aspects of our proposal that make our paper worthy of publication in Nature Communications despite the critical questions raised.

One of the main claims of the paper is that the laser pulse leads to a non-thermal (Weyl) state that has a relatively long lifetime. However, the authors do not explain why the system does not thermalize. In interacting quantum systems, thermalization can occur following a quench. Here, the non-thermal nature of the long-lived state is unclear. One can worry it is a relic of the approximations made. For instance, it is known that non-interacting (e.g. mean field states) do not thermalize.

We appreciate the reviewer's critical comment, which allowed us to improve the manuscript considerably. First of all, as outlined above in our letter, we would like to point out that nonthermality is not crucial for the general viability of our proposal. The reduction of Hubbard U alone would allow one to switch to the WSM phase even if it happened adiabatically. Of course, nonthermality is part of our claim as it extends dramatically the need of finetuning of the excitation to reach the WSM phase in practice, but one should perhaps rather see it as an added bonus.

Regarding the issue of nonthermality being a possible artefact of our approximation: Nonthermal ordering is not an artefact of mean-field theory, but has also been observed for instance in dynamical mean-field calculations and with other methods beyond mean-field, cf. Ref. 25 (Tsuji et al.) and references therein. Nonthermality becomes exact for infinite time scales only in the integrable limit (e.g. in static mean field), but survives as prethermalization even for non-integrable systems, with divergent time scales when approaching a nonthermal critical point. Therefore, we expect that these time scales can be large, and the main issue regarding time scales for thermalization and relaxation back out of the WSM phase are not correlation effects, but rather coupling to phonons, as discussed in the manuscript.

We have amended the manuscript accordingly and discuss these aspects more clearly now.

- Weyl semimetals have been observed experimentally in equilibrium, where they are easier to probe. Obtaining them via a non-equilibrium pathway is thus not as exciting. Combined with the fact that time-resolved ARPES measurements will be difficult, I do not expect that many experimentalists will attempt to follow this numerical recipe. If the non-equilibrium state had intrinsically new properties, it would be more exciting, but this is not discussed in the manuscript.

We thank the reviewer for pointing out his/her concerns regarding the difficulties of our approach. However, we would like to view this as a complementary approach to possible equilibrium routes towards AF-Weyl states. We would like to leave it open to the rapidly

growing and evolving experimental pump-probe community to follow our recipe, which does not solely rely on the appearance of nonthermal states as pointed out above.

Moreover, we believe that the novelty of our approach is three-fold:

(i) As the reviewer confirms, there is no convincing evidence yet of magnetic Weyl materials in pyrochlores, despite intense searches for many years now. The unambiguously observed equilibrium Weyl states are all for nonmagnetic materials to the best of our knowledge.

(ii) Nonequilibrium states, while harder to observe, have the advantage that they are controlled by ultrashort laser pulses and thus allow for ultrafast optical switching, which equilibrium states do not. This opens an entirely new research area in combination with topology and magnetism, for which the current proposal should be seen as a first step rather than a definite answer to all open questions.

(iii) Our proposal shows that light-induced states that live longer than the laser pulse are possible in principle. Actual lifetimes of such states of course do depend on the "drastic" approximations involved to render the problem numerically tractable at all. But as laid out above and discussed more carefully in the revised manuscript, the longevity of magnetic Weyl states beyond the duration of the laser pulse are expected to hold even beyond those approximations.

- In the numerical simulations, how is the laser pulse coupled to the electronic degrees of freedom? Also, can the authors explain the choice of polarization? Would a circularly polarized beam lead to significant differences?

In our simulation, the laser is described by a time-dependent vector potential. This vector potential is coupled to the generalized Kohn-Sham (TDDFT+U) Hamiltonian via the minimal coupling prescription.

As the system exhibits a groundstate which is the all-in all-out configuration, we decided to use the z direction for the laser polarization, which is not aligned with any of the local magnetic moments of the Ir atoms. Moreover, this corresponds to an experimentally meaningful situation of a laser at normal incidence on a crystal cut along the [100] crystallographic direction.

A circularly polarized beam would couple only to specific components of the spinor states, and break time-reversal symmetry explicitly, which is already spontaneously broken by the magnetic ordering in the material. In some cases this has been shown to affect magnetic ordering perpendicular to the plane of circular polarization via an effective magnetic field (see Ref. 4). These effects require further analysis beyond the scope of the present work. However, we expect the main effect at work in our proposal to still hold true, namely that screening effects and reduction of onsite U would partially quench the magnetic ordering also for circular polarization.

Following the reviewer's helpful suggestion, we have added a sentence on the minimal coupling prescription for the laser in the Methods section.

- Why is the AFI phase referred to as a Mott phase?

This is indeed an incorrect name erroneously used once in the manuscript. We have amended this and renamed the phase as AFI throughout in the revised manuscript.

- The double-time lesser Green's function is mentioned without explanation in the text. This is not appropriate for the broad readership of Nature Communications. The authors should add a heuristic explanation.

Very good point, we have followed the reviewer's suggestion and added a heuristic explanation of the Green's function for the broad readership of Nature Communications.

- On page 4, the authors refer to a DFT calculation for the band gap, but not provide a reference.

This is in fact a comparison with our own DFT result for the band gap. We have changed the sentence in the manuscript to remove this confusion: "Our choice of parameters is motivated by comparing with the size of the band gap from our density functional theory calculation."

- In Fig. 3, what causes the broadening of the spectral lines?

The line broadening is due to energy-time uncertainty limited energy resolution due to finite probe pulse duration in the time-resolved spectra, which we now explain in the revised manuscript.

- On page 2, the authors use reference 11 to mention interesting equilibrium phases predicted for the pyrochlore iridates. However, the following reference has appeared prior to 11:

Pesin & Balents, Mott physics and band topology in materials with strong spin-orbit interaction, Nature Physics volume 6, pages 376–381 (2010)

This is correct, the reference has been added as a first reference for the equilibrium phases.

- On page 3, references 26-27 are used regarding the origin of magnet order in the iridates. The earlier reference 11 seems more appropriate, as well as the subsequent review article: Witczak-Krempa, Chen, Kim, Balents, Correlated quantum phenomena in the strong spin-orbit regime, Annual Review of Condensed Matter Physics, Vol. 5: 57-82 (2014)

This review seems to also summarize the physics encoded in the Hubbard model used by the authors.

We thank the reviewer for pointing this out. We have changed the manuscript and replaced the mentioned references as suggested.

Reviewer #2 (Remarks to the Author):

This manuscript theoretically proposes a way to realize the Weyl semimetal along with related magnetism in pyrochlore iridates. For that the authors evoke a ultrafast, nonequilibrium modification of the electron interaction by laser pulses. The proposal is interesting as a nonequilibrium modification of material properties with the iridates being of currently much studied one. So I basically recommend its publication, but I have several comments.

We thank the reviewer for his/her very positive assessment of our work.

1. The authors heavily rely on the (DFT+U) formalism. For a description of real materials the method is often employed, but at the same time it is well known that the method has some ambiguity due to double counting terms. So the authors should discuss how this may affect the treatment, along with its relevance to t -dependent (relaxation) phenomena. Alternative method could be t -dependent DMFT, and they might like to comment on this, too.

In our work we employed the fully-localized limit (FLL) double counting. It is true that the double counting has been shown to influence the results in case of spin-orbit coupling, see for instance Ylvisaker et al., PRB 79, 035103 (2009).

This ambiguity is also present in other methods such as the constrained random phase approximation (cRPA), or density functional theory plus (extended) dynamical mean-field theory (DFT+(E)DMFT). However, this double counting problem should affect equally the ground-state and the time-dependent calculations.

Based on the fact that our magnetic ground state seems to agree well with other studies, we believe that we are capturing correctly the physics at place.

The other option of time-dependent DMFT is of course a relevant suggestion in the case of the model calculation. However, DMFT is rather difficult to apply at this point to the nonequilibrium 3D multi-band case under consideration. For the static case, DMFT would change the phase diagram mainly quantitatively, but not qualitatively unless one is interested in the Slater-to-Mott crossover for the AFI phase. For the driven case, t -DMFT is also expected to give qualitatively similar behavior (see reference 25, Tsuji et al.) to the Hartree-Fock case, in particular nonthermal magnetic order and nonthermal criticality. Finally, the main cause of relaxation dynamics and thermalization for possible experiments motivated by our idea, namely coupling to low-energy phonons, would not be captured by a purely electronic t -DMFT calculation either.

2. The authors say that laser-modified states in weakly-interacting systems are distinct from those for ordered phases for correlated electrons with slow and often nonthermal dynamics[22,23]. However, already in the original proposal of the Floquet topological insulator[7] and subsequent papers [Dehghani et al, PRB 91, 155422 (2015)], nonthermal distribution in Floquet states are shown to be important, so the statement can be misleading. Ref[22], by the way, treats an electron-phonon system where phonon relaxation can be involved.

Very good point. Nonthermal distribution in Floquet states are indeed important for instance for the Hall effect in Floquet Chern insulators, but they do not affect Floquet band structures immediately. This is drastically different in our case, since the order parameter depends on distributions, and the bands in turn depend on the order parameter. We have changed the sentences in the introduction to reflect these points more clearly and added the Dehghani et al. reference.

Rests are some points on citation.

3. When the authors say that "Floquet topological states" ... captured by driven noninteracting models, ref[7] may be cited as a first example.

We have followed this suggestion.

4. The authors stress the distinction between the existing Floquet states and the present proposal of the electron interaction modification. That is a good point, but still they can refer to eg Floquet engineering of strongly-correlated magnets into chiral spin states [Kitamura et al, PRB 96, 014406 (2017)] as a related subject.

This is a good point and we have added this reference accordingly.

===

Summary of major changes, highlighted in red in the revised manuscript:

- changed "nonthermal" to "nonequilibrium" in title and abstract
- expanded discussion of Floquet occupations and chiral magnetic states in the introduction
- extended discussion of nonthermal behavior beyond mean-field approximation
- heuristic explanation of lesser Green's function
- explanation of line broadening as a result of time-energy uncertainty
- clarification of aspects of nonthermality in the summary

Reviewers' comments:

Reviewer #1 (Remarks to the Author):

The authors have improved the manuscript following the first round of reports. In particular, my concerns have been largely addressed. However, the authors should address the following important question before I can make my decision.

Q: On pages 7 and 8, it is stated:

"While our mean-field model falls into the integrable class, thermalization is slowed down considerably even in nonintegrable systems close to a nonthermal critical point, at which the thermalization time diverges. Therefore nonthermality on time scales that exceed the time scale of the laser perturbation is not an artefact of the mean-field approximation employed in the present work, but is expected to survive when true correlation effects are included."

In the model/protocol under consideration, what exactly is the nonthermal critical point that the authors allude to? In equilibrium, the AFI-to-WSM transition in Fig.1b is first order. Do the authors claim that such a transition becomes "critical" out of equilibrium? A clarification is in order.

On page 8, the authors also mention:

"Nonthermal order in our model calculations is furthermore consistent with our TDDFT+U results (Figure 2b), which do not involve a mean-field approximation. This suggests that our finding is generic, does not require finetuning, and should thus be observable experimentally."

However, my understanding is that such methods are actually more sophisticated mean-field theories, where one is left with a sharp bandstructure at the end of the day. This is in agreement with what the authors state in the appendix:

"magnetization are computed by propagating the generalized Kohn-Sham equations within time-dependent density functional theory including mean-field interactions (TDDFT+U), as provided by the Octopus package, using the ACBNO functional together with the local-density approximation (LDA)"

As such, this undermines the claim that the long prethermal regime is generic. Can the authors explain this important point?

A smaller comment:

Fig.1a could be made clearer. Indeed, the gray area on top could be removed and the magnetic order could appear directly in the relevant regions of the phase diagram, namely AFI and WSM.

Reviewer #2 (Remarks to the Author):

In the revised manuscript, the comments raised by this reviewer have been properly taken into account. So the paper can now be accepted for publication in this journal.

Reviewers' comments:

Reviewer #1 (Remarks to the Author):

The authors have improved the manuscript following the first round of reports. In particular, my concerns have been largely addressed. However, the authors should address the following important question before I can make my decision.

Q: On pages 7 and 8, it is stated:

“While our mean-field model falls into the integrable class, thermalization is slowed down considerably even in nonintegrable systems close to a nonthermal critical point, at which the thermalization time diverges. Therefore nonthermality on time scales that exceed the time scale of the laser perturbation is not an artefact of the mean-field approximation employed in the present work, but is expected to survive when true correlation effects are included.”

In the model/protocol under consideration, what exactly is the nonthermal critical point that the authors allude to? In equilibrium, the AFI-to-WSM transition in Fig.1b is first order. Do the authors claim that such a transition becomes “critical” out of equilibrium? A clarification is in order.

We thank the reviewer for this important question. Indeed the critical point that we are referring to is not the AFI-to-WSM transition but rather the WSM-to-metal magnetic phase transition, which is second order. Please compare with the nonequilibrium DMFT study by Tsuji et al., Ref. 25. In their study there is no first order AFI-to-WSM transition but just an AFI to normal-metal second order transition, which brings about a nonthermal critical point.

On page 8, the authors also mention:

“Nonthermal order in our model calculations is furthermore consistent with our TDDFT+U results (Figure 2b), which do not involve a mean-field approximation. This suggests that our finding is generic, does not require finetuning, and should thus be observable experimentally.”

However, my understanding is that such methods are actually more sophisticated mean-field theories, where one is left with a sharp bandstructure at the end of the day. This is in agreement with what the authors state in the appendix:

“magnetization are computed by propagating the generalized Kohn-Sham equations within time-dependent density functional theory including mean-field interactions (TDDFT+U), as provided by the Octopus package, using the ACBN0 functional together with the local-density approximation (LDA)”

As such, this undermines the claim that the long prethermal regime is generic. Can the authors explain this important point?

We agree with the referee that the statement "do not involve a mean-field approximation" could be understood in the wrong way and is perhaps not an entirely faithful claim at this point. Since the question "how correlated is the ACBN0 functional" is beyond the scope of our work and should be answered in independent studies, we therefore remove this statement from our present work.

Crucially though, this does in no way undermine our main punchline, as already highlighted by us in the previous rebuttal round. Once again: (i) the main claim of our paper is not nonthermality, but light-induced reduction of U leading to WSM even for instantaneously thermalized states; (ii) if nonthermality were to occur, which is actually expected within mean-field and even beyond mean-field in long-range ordered phases near phase transitions, then the range in phase space on which light-induced WSM could be observed even grows beyond the range for which it would happen for thermal states.

Moreover, we have now included coupling to a thermal heat bath that leads to thermalization of the system (both of the global energy density and of the magnetization) on a "typical" thermalization time scale that would still render the nonequilibrium WSM observable with state-of-the-art pump-probe spectroscopies.

A smaller comment:

Fig.1a could be made clearer. Indeed, the gray area on top could be removed and the magnetic order could appear directly in the relevant regions of the phase diagram, namely AFI and WSM.

We have taken this point into account and thank the reviewer for the useful suggestion.

Reviewer #2 (Remarks to the Author):

In the revised manuscript, the comments raised by this reviewer have been properly taken into account. So the paper can now be accepted for publication in this journal.

We are glad to hear the reviewer's recommendation for publication of our manuscript.

REVIEWERS' COMMENTS:

Reviewer #1 (Remarks to the Author):

The authors have addressed my comments. In addition, they have improved their analysis by including the effects of a thermal bath. This ensures the thermalization of the electronic system under study and should provide a more realistic description. I now recommend the publication of the paper in Nature Communications.